# Development and validation of a risk prediction model of preterm birth for women with preterm labour symptoms (the QUIDS study): A prospective cohort study and individual participant data meta-analysis

Sarah J. Stock[1,2]*, Margaret Horne[2], Merel Bruijn[2], Helen White[3], Kathleen A. Boyd[4], Robert Heggie[4], Lisa Wotherspoon[2], Lorna Aucott[5], Rachel K. Morris[6], Jon Dorling[7], Lesley Jackson[8], Manju Chandiramani[9], Anna L. David[10], Asma Khalil[11], Andrew Shennan[12], Gert-Jan van Baaren[13], Victoria Hodgetts-Morton[6], Tina Lavender[3], Ewoud Schuit[14], Susan Harper-Clarke[15], Ben W. Mol[16], Richard D. Riley[17], Jane E. Norman[18], John Norrie[1]

1 Usher Institute, University of Edinburgh, Edinburgh, United Kingdom, 2 MRC Centre for Reproductive Health, Queen's Medical Research Institute, University of Edinburgh, United Kingdom, 3 Faculty of Biology, Medicine and Health, University of Manchester, United Kingdom, 4 Institute of Health and Wellbeing, University of Glasgow, United Kingdom, 5 Health Services Research Unit, University of Aberdeen, United Kingdom, 6 Institute of Applied Health Research, University of Birmingham, United Kingdom, 7 IWK Health Centre, Halifax, Nova Scotia, Canada, 8 Queen Elizabeth Hospital, Glasgow, United Kingdom, 9 Guy's and St Thomas NHS Foundation Trust, London, United Kingdom, 10 Elizabeth Garrett Anderson Institute for Women's Health, University College London, United Kingdom, 11 Vascular Biology Research Centre, Molecular and Clinical Sciences Research Institute, St George's University of London, United Kingdom, 12 Department of Women and Children's Health, School of Life Course Sciences, Kings College London, United Kingdom, 13 Department of Obstetrics and Gynaecology, Amsterdam University Medical Center, Amsterdam, the Netherlands, 14 Department of Epidemiology, Julius Center for Health Sciences and Primary Care, University Medical Center Utrecht, Utrecht University, Utrecht, the Netherlands, 15 Lay Representative (no affiliation), 16 Department of Obstetrics & Gynaecology, Faculty of Medicine, Nursing and Health Sciences, Monash University, Clayton, Australia, 17 Centre for Prognosis Research, School of Medicine, Keele University, Keele, United Kingdom, 18 Faculty of Health Sciences, University of Bristol, Bristol, United Kingdom

* sarah.stock@ed.ac.uk

**Data Availability Statement:** Data is held in a public repository (University of Edinburgh

## Abstract

### Background

Timely interventions in women presenting with preterm labour can substantially improve health outcomes for preterm babies. However, establishing such a diagnosis is very challenging, as signs and symptoms of preterm labour are common and can be nonspecific. We aimed to develop and externally validate a risk prediction model using concentration of vaginal fluid fetal fibronectin (quantitative fFN), in combination with clinical risk factors, for the prediction of spontaneous preterm birth and assessed its cost-effectiveness.

### Methods and findings

Pregnant women included in the analyses were $22^{+0}$ to $34^{+6}$ weeks gestation with signs and symptoms of preterm labour. The primary outcome was spontaneous preterm birth within 7

Datashare). The DOI for data availability is https://datashare.ed.ac.uk/handle/10283/3885.

**Funding:** This publication presents independent research funded by the National Institute for Health Research (NIHR) HTA Programme (Project Number 14/32/01). All listed authors were funded via this programme. Funder website URL: https://www.nihr.ac.uk/explore-nihr/funding-programmes/health-technology-assessment.htm. The funder had no role in study design, data collection and analysis, decision to publish, or preparation of the manuscript. SJS is funded by the Wellcome Trust (209560/Z/17/Z). The funder had no role in study design, data collection and analysis, decision to publish, or preparation of the manuscript. We are grateful to HOLOGIC (Marlborough, MA, USA) for providing adapted analyzers at each site allowing quantitative fFN values to be masked from clinicians and offered training and support to each site for fFN testing. They also allowed QUIDS sites to purchase tests at the lowest list price (NHS treatment cost). HOLOGIC had no involvement in data collection, analysis or interpretation, and no role in the writing of this manuscript or the decision to submit for publication. We are also grateful to PARSAGEN DIAGNOSTICS who gifted Partosure test kits for a substudy comparing different biochemical test of preterm labour (to be reported separately) and offered training and support to sites and MEDIX BIOCHEMICA who provided test kits at a reduced cost and offered training and support to sites. These companies had no other involvement in study design, implementation, analysis or interpretation of results.

**Competing interests:** I have read the journal's policy and the authors of this manuscript have the following competing interests: SJS is a member of PLOS Medicine's Editorial Board. All authors had financial support from National Institute for Health Research (NIHR) National Institute for Healthcare Research; SJS reports financial support form the Wellcome Trust (209560/Z/17/Z), non-financial support from HOLOGIC, non-financial support from PARSAGEN, and non-financial support from MEDIX BIOCHEMICA to support the conduct of the study; JD reports grants from Nutrinia in 2017 and 2018 which were part of his salary to work as an expert advisor on a trial; MC reports that she has done advisory work for HOLOGIC unrelated to the submitted work, and has been supported by HOLOGIC to attend a conference in the last 12 months; ALD reports personal fees from HOLOGIC outside the submitted work; and salary support from the NIHR UCLH/UCL Biomedical Research Centre; AK reports grants and prediction tests from

days of quantitative fFN test. The risk prediction model was developed and internally validated in an individual participant data (IPD) meta-analysis of 5 European prospective cohort studies (2009 to 2016; 1,783 women; mean age 29.7 years; median BMI 24.8 kg/m$^2$; 67.6% White; 11.7% smokers; 51.8% nulliparous; 10.4% with multiple pregnancy; 139 [7.8%] with spontaneous preterm birth within 7 days). The model was then externally validated in a prospective cohort study in 26 United Kingdom centres (2016 to 2018; 2,924 women; mean age 28.2 years; median BMI 25.4 kg/m$^2$; 88.2% White; 21% smokers; 35.2% nulliparous; 3.5% with multiple pregnancy; 85 [2.9%] with spontaneous preterm birth within 7 days). The developed risk prediction model for spontaneous preterm birth within 7 days included quantitative fFN, current smoking, not White ethnicity, nulliparity, and multiple pregnancy. After internal validation, the optimism adjusted area under the curve was 0.89 (95% CI 0.86 to 0.92), and the optimism adjusted Nagelkerke R$^2$ was 35% (95% CI 33% to 37%). On external validation in the prospective UK cohort population, the area under the curve was 0.89 (95% CI 0.84 to 0.94), and Nagelkerke R$^2$ of 36% (95% CI: 34% to 38%). Recalibration of the model's intercept was required to ensure overall calibration-in-the-large. A calibration curve suggested close agreement between predicted and observed risks in the range of predictions 0% to 10%, but some miscalibration (underprediction) at higher risks (slope 1.24 (95% CI 1.23 to 1.26)). Despite any miscalibration, the net benefit of the model was higher than "treat all" or "treat none" strategies for thresholds up to about 15% risk. The economic analysis found the prognostic model was cost effective, compared to using qualitative fFN, at a threshold for hospital admission and treatment of $\geq$2% risk of preterm birth within 7 days. Study limitations include the limited number of participants who are not White and levels of missing data for certain variables in the development dataset.

## Conclusions

In this study, we found that a risk prediction model including vaginal fFN concentration and clinical risk factors showed promising performance in the prediction of spontaneous preterm birth within 7 days of test and has potential to inform management decisions for women with threatened preterm labour. Further evaluation of the risk prediction model in clinical practice is required to determine whether the risk prediction model improves clinical outcomes if used in practice.

## Trial registration

The study was approved by the West of Scotland Research Ethics Committee (16/WS/0068). The study was registered with ISRCTN Registry (ISRCTN 41598423) and NIHR Portfolio (CPMS: 31277).

## Author summary

### Why was this study done?

- Preterm labour is notoriously challenging for clinicians to diagnose due to the nonspecific nature of presenting signs and symptoms.

Parsagen Diagnostics, paid to the institution, outside the submitted work; AS has grants and prediction tests from HOLOGIC, paid to the institution outside the submitted work; BM reports other grants from NHMRC, personal fees from ObsEva, Merck, Guerbet, and grants from Guerbet and Merck outside the submitted work; RDR reports personal fees from Roche outside the submitted work; JEN reports personal fees from Dilafor outside the submitted work; no other relationships or activities that could appear to have influenced the submitted work. All authors have completed the ICMJE uniform disclosure form at www.icmje.org/coi_disclosure.pdf.

**Abbreviations:** AUC, area under the receiver operating characteristic curve; FDA, US Food and Drug Administration; fFN, fetal fibronectin; ICER, incremental cost-effectiveness ratio; IPD, individual participant data; NHS, National Health Service; NICE, National Institute of Healthcare Excellence; NMB, net monetary benefit; PAMG-1, placental alpha microglobulin-1; phIGFBP-1, phosphorylated insulin-like growth factor binding protein-1; QALD, quality-adjusted life day; QALY, quality-adjusted life year; TRIPOD, Transparent Reporting of a multivariable prediction model for Individual Prognosis or Diagnosis.

- The outcomes and health of preterm babies can be significantly improved by well-timed and appropriate interventions, but unnecessary overtreatment of women is common and costly.

- Qualitative fetal fibronectin (fFN) is one test used to aid the diagnosis of preterm labour.

## What did the researchers do and find?

- We developed and validated a risk prediction model to improve the prediction of impending preterm birth using a combination of qualitative fFN results and additional clinical risk factors.

- The primary outcome was spontaneous preterm birth within 7 days of fFN testing.

- We developed and internally validated the model using data from 1783 European women and externally validated the model in a prospective cohort study of 2,924 UK women. Women with signs and symptoms of preterm labour were included in the analysis between $22^{+0}$ to $34^{+6}$ weeks gestation.

- At a threshold of $\geq 2\%$ risk of birth within 7 days of testing, we found the prognostic model to be cost effective in comparison to fFN alone.

## What do these findings mean?

- Our findings indicate that the prognostic model has promising potential to improve prediction of spontaneous preterm birth.

- The model is well placed to aid management decisions and discussions with women presenting with signs and symptoms of preterm labour.

- Limitations were that there were low numbers of participants in the studies who were not White, and there were some missing data in the risk predictor development cohort.

## Introduction

Preterm birth, defined as delivery before 37 completed weeks of gestation, is the leading cause of neonatal morbidity and mortality, with nearly 15 million preterm births worldwide every year [1,2]. The majority of preterm births result from spontaneous preterm labour [3]. Timely interventions in women with preterm labour can substantially improve health outcomes for preterm babies. Such measures include birth in a centre with appropriate neonatal care facilities [4], antenatal corticosteroids for lung maturity administered within 7 days of birth [5], and peripartum magnesium sulphate for neuroprotection [6]. However, establishing a diagnosis of preterm labour is very challenging, as signs and symptoms of preterm labour are common and can be nonspecific. Large numbers of women with symptoms of preterm labour are treated unnecessarily in order to ensure appropriate treatment of the minority who actually give birth preterm [7,8]. One-third of all antenatal hospital admissions are with a suspected diagnosis of preterm labour [9], resulting in a substantial economic burden to health services [10] and

negative financial and emotional impacts on women and families [11]. Unnecessary prophylactic treatments cause adverse maternal and neonatal effects, for example, tocolytics can cause adverse drug reactions in mothers, and unnecessary antenatal corticosteroids are associated with adverse neurodevelopmental outcomes in children [12–14].

Clinical tests may help confirm or refute the diagnosis of preterm labour to help target appropriate treatment. Transvaginal ultrasound measurement of cervical length is one strategy that has moderate predictive performance, but it requires specialised equipment and highly trained staff [15]. Biochemical tests of preterm labour measure the concentration of markers of impending birth in cervicovaginal secretions, are easy to perform, and have potential for widespread implementation, including in settings where ultrasound is unavailable. The most established biochemical test is vaginal fluid fetal fibronectin (fFN; Hologic, Marlborough, Massachusetts, United States of America) [16]. This has moderate accuracy for ruling out impending preterm birth based on a single threshold of 50 ng/ml [10,16]. fFN concentration in ng/ml (quantitative fFN) may have better prognostic value than a single threshold test [10]. However, quantitative fFN does not have US Food and Drug Administration (FDA) Approval in the USA, and recent United Kingdom National Institute of Healthcare Excellence (NICE) guidance concluded that there was insufficient evidence to recommend use of quantitative fFN to determine the risk of preterm labour [15].

Combining biochemical test results with clinical risk factors in a risk prediction model can improve the predictive ability of a model [17]. We thus hypothesised that quantitative fFN could be combined with information on other known risk factors for preterm birth to better determine an individual's risk of preterm birth and thereby help inform decision-making for women with symptoms of preterm labour. We aimed to develop and externally validate a risk prediction model for preterm birth comprising these 2 elements using robust methodology.

## Methods

### Study oversight

The QUIDS study (QUIDS: Quantitative fetal fibronectin to improve decision-making in women with signs and symptoms of preterm birth) protocols were developed according to the relevant guidelines [17–19], published [20,21], and made available through the ISRCTN and study websites. The individual participant data (IPD) meta-analysis was registered on the International Prospective Register of Systematic Reviews (PROSPERO; CRD42015027590), and the main prospective cohort study (ISRCTN 41598423) was registered. The research was approved by the West of Scotland Research Ethics Committee (16/WS/0068). All participants gave written informed consent.

The findings are reported in line with the Transparent Reporting of a multivariable prediction model for Individual Prognosis or Diagnosis (TRIPOD) statement (S1 Table) [22].

### Choice of health technology

The choice to include quantitative fFN was based on it being the only quantitative biochemical test of preterm labour available, and previous research had concluded it may have potential to be a more useful predictor of preterm birth than when used as a dichotomous test [10]. Two other biochemical tests of preterm labour are also marketed in some countries—Actim Partus (Medix Biochemica, Espoo, Finland), which measures phosphorylated insulin-like growth factor binding protein-1 (phIGFBP-1), and Partosure (Parsagen Diagnostics, Boston, Massachusetts, USA), which measures placental alpha microglobulin-1 (PAMG-1); but evidence supporting their use is currently limited [15]. We performed an exploratory substudy comparing test performance that will be published separately.

## Model outcome, target population, and candidate predictors

We included in the analyses pregnant women who were $22^{+0}$ to $34^{+6}$ weeks gestation with signs and symptoms of preterm labour (as defined by investigators), intact membranes, and no contraindication to fFN testing (Table A in S1 Text). The primary outcome was spontaneous preterm birth within 7 days of quantitative fFN test, which was a binary outcome. The choice of this outcome was determined by our previous qualitative study of women and clinicians [23].

We prespecified candidate predictors (i.e., those, considered to have potential to influence the risk of spontaneous preterm birth before model development). These are shown in Table B in S1 Text. Only maternal age, BMI, ethnicity, smoking, nulliparity, multiple pregnancy, gestational age at assessment, previous spontaneous preterm birth before 34 weeks, cervical length, and fFN were available in each study, and, therefore, we included these 10 candidate predictors in the model development. All included candidate predictors are available around the time of initial assessment for signs and symptoms of preterm birth. Tocolysis was included in sensitivity analysis to explore any potential treatment effect on delaying birth.

## Predictive performance measures

After model development, we examined the predictive performance of the model's predicted risks both internally (using the development data) and externally (using new data), in terms of discrimination and calibration. We summarised the discrimination performance using the area under the receiver operating characteristic curve (AUC) (with 0.5 representing no discrimination beyond chance and 1 perfect discrimination). We visualised calibration of predicted and observed risks using a calibration plot including a nonparametric (loess smoother) calibration curve and the ratio of predicted (expected) to observed risk for each 10th of predicted risk. We quantified calibration across all participants by the calibration slope (ideal value 1), calibration-in-the-large (ideal value 0), and expected to observed ratio (ideal value 1). We expressed overall model fit using Nagelkerke $R^2$.

## Model development

The IPD meta-analysis included prospective cohort studies or randomised control trials of women with signs and symptoms of preterm labour that included quantitative fFN results determined by 10Q rapid fFN analyser system and pregnancy outcome data. Methods to identify studies are described in the published protocol [21]. In brief, prior to applying for funding for the study (April 2014), we performed a literature search for completed and ongoing cohort studies of quantitative fFN using search terms for quantitative fFN and preterm birth, including databases, clinical trial registries, general search engines, and systematic reviews, and consulted preterm birth researchers and networks and the manufacturers of quantitative fFN (Hologic) to help ensure capture of all relevant studies. Study manuscripts and/or protocols were screened by 2 researchers. We identified 10 studies of quantitative fFN that were potentially eligible. Four early datasets (in 3 manuscripts) used ELISA to determine the concentration of fFN and were excluded as the different method of analysis and earlier period of study would increase heterogeneity [24,25,26]. Therefore, 6 studies fulfilled the eligibility criteria, of which 5 could provide data. Details and assessment of bias of included studies are in Table C in S1 Text and Table D in S1 Text.

Data were cleaned prior to analysis. Multiple imputation of missing values by chained equations was performed within each original study separately [27]. Sixty imputations were performed, and Rubin's rules used to combine estimates across datasets.

We assessed heterogeneity of predictor effects across the studies providing IPD using a 2-stage approach, fitting a model with all candidate predictors in each participating study, and then performing a random-effects meta-analysis of each predictor's adjusted effect. We

quantified heterogeneity of the predictor effects using estimated between-study variance ($\tau^2$) and $I^2$ (Table E in S1 Text).

We used logistic regression to develop models. Our intention was to include predictors that were available in each study (Table B in S1 Text.) and that had low heterogeneity of their adjusted effect across studies. Although cervical length was included in all studies, there were high levels (85%) of missing data in 2 of the studies. This reflects the fact that cervical length measurement in women with symptoms of preterm labour requires 24-hour availability of highly skilled staff and ultrasound equipment and is not universally available. We therefore excluded cervical length from our initial model but included it in an alternative model (with data from the 3 studies with near-complete observations on cervical length) to explore whether it had potential to improve discrimination of the quantitative fFN-based risk prediction model. We used a one-stage IPD meta-analysis framework with separate intercept term per study to account for clustering. We used backwards selection based on an Akaike's information criterion and a $p$-value of $<0\cdot15$ (also a full model was considered; see later). Continuous predictors were analysed on their continuous scale. Nonlinear terms were identified using multivariable fractional polynomials. Apparent model performance was assessed in the same data used for model development.

### Internal validation

We performed internal validation using nonparametric bootstrap resampling with 100 bootstrap samples (replicating all steps of our model development), in particular to estimate optimism due to overfitting. For each performance measure, the differences between the bootstrap model's apparent value in each bootstrap sample and its test value in the original dataset were averaged to give the estimated optimism. We then subtracted this optimism from apparent performance estimates of the original model to give optimism-adjusted estimates. We used the optimism-adjusted calibration slope as a uniform shrinkage factor to shrink the beta coefficients of the original model to adjust for overfitting. Subsequently, we reestimated study-specific intercept terms while fixing the shrunken beta coefficients (via an offset term) to ensure that overall calibration-in-the-large was maintained in each study separately.

### External validation

We performed a prospective cohort study of women in 26 consultant-led obstetric units in the UK between June 2016 and October 2018 to externally validate the developed risk prediction model, in accordance with a published protocol [20]. Hospitals had different levels of neonatal care facilities in both rural and urban settings (Table F in S1 Text).

Women with signs and symptoms of preterm labour (any or all of back pain, abdominal cramping, abdominal pain, light vaginal bleeding, vaginal pressure, uterine tightenings or contractions), in whom admission/treatment were being considered, were identified on presentation to obstetric services. Informed consent was invited before speculum examination. This approach meant that vaginal fluid fFN samples were collected at routine speculum examination, as they are in clinical practice. However, certain exclusion criteria could only be applied at speculum examination (for example, vaginal bleeding or evidence of ruptured membranes), so a proportion of women were not eligible for fFN testing after consent was given. These data are reported but not included in analysis. Further details of cost-effectiveness and acceptability of the model and experiences of clinicians and women involved in the study will be reported elsewhere.

Samples for fFN analysis were taken with an fFN specimen collection kit as per manufacturer's instructions. The sample was run at a near bedside Rapid fFN 10Q analyser specially adapted for the study. The quantitative fFN result was masked from clinicians and stored as a

3-letter code. The qualitative fFN result (positive/negative/invalid based on a 50-ng/ml thresh-old) was displayed for clinicians to base clinical decision-making on, according to local protocols.

Samples were run as per manufacturer's instructions (Table G in S1 Text) with appropriate quality control measures (Table H in S1 Text). Screening data and data about quantitative fFN testing were collected on paper-based case report forms, which were then inputted onto a web-based electronic database by research staff. All other data were collected from the participant records and recorded on the study database.

The study database was locked on 14 January 2019. Data were cleaned, and multiple impu-tation of missing data performed as described above (using 8 imputations). A random-effects meta-analysis was used to combine the 5 study-specific intercepts included in the developed prediction model, and the pooled estimate was then used as the intercept when applying the model to individuals in the prospective cohort data. Measures of the model's performance were then assessed using similar methodology to that described above. Performance was also examined after recalibration-in-the-large, which uses the external validation data to update the original model's intercept while holding predictor effects at their original value.

### Additional models and subgroup analyses

During model development in the IPD meta-analysis, we also examined the inclusion of all predictors within the model (i.e., without variable selection; this was not prespecified) and explored the inclusion of cervical length (as described above in *Model Development*). A further investigation included a model with tocolysis as a categorical variable (administered/not administered) to explore any potential treatment effect of tocolysis in delaying birth (results shown in Table I in S1 Text).

Additional analyses that were not prespecified were undertaken in the external validation cohort to test the robustness of our model and included validating the model (i) with an outcome of all preterm births within 7 days (i.e., including spontaneous and provider initiated preterm births, as opposed to only spontaneous preterm births); (ii) in singletons only (as opposed to including twins); and (iii) in a complete-case analysis (as opposed to one with imputation of miss-ing data). We had intended to do a further analysis with all fFN test results included (as opposed to including only the first recorded fFN result of each study participant); however, the number of women who had more than one fFN test was too low to make this robust (206; 7.0%).

### Net benefit

In order to evaluate the potential clinical value of using the risk prediction model to inform decision-making, we performed net benefit analyses (which was not prespecified), where bene-fits and harms of a risk prediction model are put on the same scale to allow direct comparison [28]. We plotted the standardised net benefit across a range of predicted risks of preterm birth within 7 days (0% to 20%). For the purpose of this analysis, we assumed that the standard of care for this population with signs and symptoms of preterm labour was to "treat" (e.g., admit to an appropriate hospital, give antenatal corticosteroids and magnesium sulphate) and that a percentage predicted risk of preterm birth would be used for "ruling out" treatment.

### Sample size calculations

The sample size for model development was fixed by the number of women participating in the existing 5 studies (1,783 women; 139 events); the number of predictors relative to the num-ber of events was about 14, and the data met minimum sample size criteria for model develop-ment [29]. For external validation, our initial sample size calculation at the cohort study outset

(1,602) was based on an anticipated event proportion of between 6% and 12%. However, it quickly became apparent that our event proportion was lower than estimated (stabilising at around 3%). Further, guidance emerged recommending that a minimum of around 100 events and 100 nonevents were needed for risk prediction model validation [30]. We therefore revised our sample size calculation during the study, aiming for 3,000 participants, to obtain approximately 100 events of preterm birth within 7 days of testing.

## Cost-effectiveness analysis

We undertook an economic evaluation from the perspective of the UK National Health Service (NHS) and Prescribed Specialised Services. The analysis was undertaken for cost year 2017/2018 and was conducted according to best practice guidelines [31,32]. The mean resource use for each participant was estimated using data from the prospective cohort study and reference unit costs (Table J in S1 Text). We performed multivariable analysis of the difference in arithmetic mean of the cost [33], estimating a number of generalised linear models with different families and link functions. The choice of final model was based on goodness of fit using the modified Park test. We calculated the percentage (0% to 100%) probability of spontaneous preterm birth with 7 days for each participant in the cohort study using the externally validated QUIDS risk prediction model. Qualitative fFN results were categorised as positive (high risk) or negative (low risk) based on a single threshold of 50 ng/ml. Table K in S1 Text provides all model parameters used in the cost-effectiveness analysis.

The base case cost-effectiveness analysis compared the cohort study results for the QUIDS risk prediction model versus a treat-all approach (recommended by NICE at gestations of less than 30 weeks gestation [16]) and versus use of qualitative fFN alone (the most commonly used test of preterm labour in the UK [34], recommended by NICE as an option for women with signs and symptoms of preterm labour over 30 weeks gestation [16]), reporting the incremental cost per correct prognosis, incremental cost per quality-adjusted life days (QALDs) gained and incremental net monetary benefit (NMB), using a willingness to pay per QALD of £54.79, equivalent to the UK threshold £20,000 per quality-adjusted life year (QALY).

The decision model was based on a decision tree framework (Fig A in S1 Text). In order to capture the longer-term cost and health outcomes associated with infants who did and did not receive treatment, we extrapolated the 7-day outcomes over a lifetime horizon. For the lifetime horizon, QALYs, rather than QALDs, are presented. Key assumptions for the models are detailed in Table L in S1 Text.

## Software

Data were cleaned in SPSS version 24. All statistical analyses were performed in R (version 3.6.1). All cost-effectiveness analyses were performed using STATA (version 19).

## Results

### Model development and internal validation

We meta-analysed the IPD from 5 clinical studies of quantitative fFN to create a risk prediction model for spontaneous preterm birth within 7 days of fFN test [20,21]. Details of the IPD meta-analysis cohort are shown in Table 1 (characteristics after multiple imputation of missing data are shown in Table M in S1 Text). The IPD contained a total of 1,783 women with signs and symptoms of preterm labour recruited in the 5 studies between 2009 and 2016, of which 139 women had spontaneous preterm birth (7.8%) within 7 days of fFN test.

We included 10 prespecified candidate predictors with potential to influence the risk of spontaneous preterm birth in the model (Table B in S1 Text) [20,21]. After backwards selection, predictors that remained in the QUIDS risk prediction model were quantitative fFN, smoking, ethnicity, nulliparity, and multiple pregnancy (Table 2). Of note, gestational age did not influence the probability of spontaneous preterm birth within 7 days (Table N in S1 Text). Quantitative fFN was transformed (square root) because of nonlinearity. The apparent discrimination performance of the model was AUC of 0.89 (95% CI 0.86 to 0.92), with Nagelkerke $R^2$ being 36% (95% CI 34% to 37%). Nonparametric bootstrap resampling resulted in a uniform shrinkage factor of 0.94, which was applied to the model's predicted effects to adjust for this small overfitting. The optimism adjusted AUC was 0.89 (95% CI 0.86 to 0.92), and the optimism-adjusted Nagelkerke $R^2$ was 35% (95% CI 33% to 37%).

We compared the performance of the developed model to that of a model including all potential predictors (i.e., without variable selection) and a model with variable selection that included cervical length (which requires specialised staff and equipment to perform). The discrimination of these additional models was similar to the principal model developed, with model performance after internal validation showing small improvement in AUCs of 0.90 (95% CI 0.88 to 0.93) and 0.92 (95% CI 0.89 to 0.94), respectively (full models shown in Table I in S1 Text), and net benefit analyses showed the models with and without cervical length to have very similar potential utility (Fig B in S1 Text). We therefore chose the original and most parsimonious model (the one with fewest predictors, which we call the QUIDS risk prediction model) for further validation.

## External validation

We performed a prospective cohort study of women in 26 consultant-led obstetric units in the UK between June 2016 and October 2018, for external validation of the QUIDS model [20]. The study flowchart (Fig 1) shows that 27 of 2,968 (0.91%) participants were ineligible at speculum examination or unable to have fFN testing completed. In addition, 17 of 2,941 (0.58%) fFN tests had an invalid result. A total of 2,924 women were included in the final analysis dataset.

The baseline characteristics of participants are shown in Table 1 (characteristics after imputation are in Table M in S1 Text). There were 85 events of spontaneous preterm birth within 7 days of fFN test in 2,924 women (2.9%).

On external validation of the QUIDS model (Table 3), the AUC remained as 0.89 (0.84 to 0.94), and Nagelkerke $R^2$ was 36% (95% CI 34% to 38%). Calibration-in-the-large was 0.26 (95% CI 0.25 to 0.28), and so we updated the intercept of the QUIDS model by recalibrating the intercept to ensure perfect calibration-in-the-large for this UK population (Predicted risk [Expected risk or "E"]/Observed cases ["O"] = 1) (Fig 2). Subsequently, the calibration curve suggested close agreement between predicted and observed risks in the range of predictions 0% to 10% (Fig 2), but some miscalibration (underprediction) at higher risks (slope 1.24 (95% CI 1.23 to 1.26)).

Model performance was similar across a number of additional analyses (Table P in S1 Text) validating the model (i) with an outcome of all preterm births within 7 days (i.e., including spontaneous and medically indicated preterm births, as opposed to only spontaneous preterm births); (ii) in singletons only (as opposed to including twins); and (iii) in a complete-case analysis (as opposed to one with imputation of missing data).

## Net benefit

In order to evaluate the potential clinical value of the risk prediction model to inform decision-making, we performed net benefit analyses, where benefits and harms of a risk prediction

**Table 1. Baseline characteristics of the QUIDS IPD meta-analysis dataset and QUIDS prospective cohort.**

| | Individual participant meta-analysis dataset (model development) n = 1,783 | | | | | | Prospective cohort (external validation) n = 2,924 |
|---|---|---|---|---|---|---|---|
| | Apostel-1 Bruijn et al. [35] | Eufis Bruijn et al. [36] | EQUIPP Abbott et al. [37] | QFCAPS Khalil et al. | UCLH/Whit David et al. | Total | |
| | n = 528 | n = 455 | n = 452 | n = 86 | n = 262 | n = 1,783 | |
| **Age** (mean [SD] years) | 29.4 (5.3) | 29.5 (5.2) | 29.5 (6.0) | 30.0 (6.1) | 31.0 (6.1) | 29.7 (5.6) | 28.2 (5.7) |
| Missing n (%) | - | - | - | - | - | - | 31 (1.1) |
| **BMI** (median [IQR] kg/m²) | 25.6 (23.5–28.1) | 25.7 (23.1–28.7) | 24.0 (21.2–28.8) | 24.6 (21.2–28.1) | 23.0 (21.0–27.8) | 24.8 (22.0–28.4) | 25.4 (22.2–30.2) |
| Missing n (%) | 303 (57.4) | 202 (44.4) | 2 (0.4) | 0 | 22 (8.4) | 529 (29.7) | 51 (1.7) |
| **Ethnicity** | | | | | | | |
| White | 342 (64.8) | 352 (77.4) | 226 (50.0) | 58 (67.4) | 145 (55.3) | 1,123 (63.0) | 2,544 (87.0) |
| South Asian | 8 (1.5) | 4 (0.9) | 25 (5.5) | 6 (7.0) | 30 (11.5) | 73 (4.1) | 165 (5.6) |
| East Asian | 6 (1.1) | 12 (2.6) | 10 (2.2) | 3 (3.5) | 12 (4.6) | 43 (2.4) | 7 (0.2) |
| African, Caribbean, Middle East | 63 (11.9) | 69 (15.2) | 159 (35.2) | 15 (17.4) | 57 (21.8) | 363 (20.4) | 98 (3.4) |
| Other | 23 (4.3) | 6 (1.3) | 32 (7.1) | 4 (4.7) | 2 (0.8) | 67 (3.8) | 68 (2.3) |
| Missing n (%) | 86 (16.3) | 12 (2.6) | 0.0 | 0.0 | 16 (6.1) | 114 (6.4) | 42 (1.4) |
| **Currently smoking** n (%) | 71 (13.4) | 41 (9.0) | 58 (12.8) | 11 (12.8) | 9 (3.4) | 190 (10.7) | 608 (20.8) |
| Missing n (%) | 43 (8.1) | 42 (9.2) | 3 (0.7) | 0.0 | 21 (8.0) | 109 (6.1) | 40 (1.4) |
| **Nulliparity** n (%) | 288 (54.5) | 262 (57.6) | 200 (44.2) | 34 (39.5) | 140 (53.4) | 924 (51.8) | 951 (32.5) |
| Missing n (%) | - | - | - | - | - | - | 171 (5.8) |
| Multiple pregnancy n (%) | 85 (16.1) | 67 (14.7) | 20 (4.4) | 0 (0) | 14 (5.3) | 186 (10.4) | 99 (3.4) |
| Missing n (%) | - | - | - | - | - | - | 29 (1.0) |
| **Gestational age** (median [IQR] weeks) | 29.4 (26.8–31.3) | 29.6 (26.7–31.6) | 29.2 (25.6–32.3) | 29.9 (27.3–33.0) | 29.0 (25.6–32.1) | 29.4 (26.4–31.7) | 31.0 (27.9–33.1) |
| Missing n (%) | | | | | | | 10 (0.3) |
| **Previous spontaneous preterm birth** <34 weeks n (%) | 69 (13.1) | 39 (8.6) | 68 (15.0) | 7 (8.1) | 13 (5.0) | 196 (11.0) | 121 (4.1) |
| Missing n (%) | - | - | - | - | - | - | 179 (6.1) |
| **Cervical length** (mean [SD] mm) | 25.0 (12.3) | 21.3 (9.5) | 26.9 (14.0) | 29.8 (9.0) | 14.2 (7.0) | 23.8 (11.5) | |
| Missing n (%) | 0.0 | 0.0 | 343 (75.9) | 0.0 | 224 (85.5) | 567 (31.8) | |
| **Qualitative fFN positive** n (%) | 199 (37.7) | 197 (43.3) | 105 (23.2) | 12 (14.0) | 35 (13.4) | 548 (30.7) | 413 (14.1) |
| Missing n (%) | | | | | | | 1 (<0.1) |
| **Quantitative fFN** (median [IQR] ng/ml) | 17.0 (4.0–112.5) | 34 (8.0–217) | 7.0 (3.0–43.8) | 4.0 (2.0–11.3) | 4.0 (2.0–16.3) | 11.0 (3.0–79.0) | 7 (4–22) |
| Missing n (%) | | | | | | | 2 (<0.1) |
| **Tocolysis** n (%) | 345 (65.3) | 319 (70.1) | 36 (8) | 7 (8) | 10 (3.8) | 717 (40.2) | 165 (5.6) |
| Missing n (%) | | | | | | | 78 (2·7) |
| **Outcome** | | | | | | | |
| Spontaneous preterm birth <7 days n (%) | 70 (13.3) | 48 (10.5) | 14 (3.1) | 2 (2.3) | 5 (1.9) | 139 (7.8) | 85 (2.9) |

Baseline characteristics of participants in QUIDS IPD meta-analysis datasets and participants in the QUIDS prospective cohort study.

fFN, fetal fibronectin; IPD, individual participant data; IQR, interquartile range; SD, standard deviation.

**Table 2. QUIDS risk prediction model for the prediction of spontaneous preterm birth within 7 days of quantitative fFN test developed (A) and internally validated (B) by IPD meta-analysis.**

| | Original model | Optimism-adjusted model |
|---|---|---|
| Study 1 intercept (95% CI) [35] | −4.828 (−6.435−−3.221) | −4.556 (−6.164−−2.949) |
| Study 2 intercept (95% CI) [36] | −5.524 (−7.131−−3.917) | −5.213 (−6.820−−3.606) |
| Study 3 intercept (95% CI) [37] | −5.937 (−7.628−−4.246) | −5.603 (−7.295−−3.912) |
| Study 4 intercept (95% CI) (QFCAPS) | −5.692 (−7.819−−3.565) | −5.372 (−7.498−−3.245) |
| Study 5 intercept (95% CI) (UCLH/Whit) | −6.149 (−8.046−−4.252) | −5.803 (−7.700−−3.906) |
| **Meta-analysed intercept** | | −5.263 (−6.046−−4.479) |
| **qfFN** | | |
| Beta coefficient | 2.029 | 1.915 |
| OR (95% CI) | 7.61 (5.68–10.19) | 6.79 (5.07–9.09) |
| **Smoking** | | |
| Beta coefficient | −0.728 | −0.687 |
| OR (95% CI) | 0.48 (0.23–1.03) | 0.50 (0.24–1.07) |
| **Not White ethnicity** | | |
| Beta coefficient | −0.147 | −0.139 |
| OR (95% CI) | 0.86 (0.50–1.48) | 0.87 (0.51–1.49) |
| **Nulliparity** | | |
| Beta coefficient | 0.391 | 0.370 |
| OR (95% CI) | 1.48 (0.95–2.30) | 1.45 (0.93–2.25) |
| **Multiple pregnancy** | | |
| Beta coefficient | 0.884 | 0.834 |
| OR (95% CI) | 2.42 (1.42–4.11) | 2.30 (1.36–3.91) |
| **Model performance** | | |
| Nagelkerke $R^2$ | 36% (34%–37%) | 35% (33%–37%) |
| AUC | 0.89 (0.86–0.92) | 0.89 (0.86–0.92) |

The QUIDS risk prediction model for the prediction of spontaneous preterm birth within 7 days of quantitative fFN test in women with signs and symptoms of preterm labour. Column A shows the baseline model and column B the model after internal validation and adjustment for optimism.

AUC, area under the curve; CI, confidence interval; E/O, expected/observed; fFN, fetal fibronectin; IPD, individual participant data; OR, odds ratio; qfFN, quantitative fetal fibronectin $((qfFN+1)/100)^{0.5}$.

Apparent model performance (A) is summarised using the means from the pooled imputed datasets for simplicity.

model are put on the same scale to allow direct comparison [28]. Fig 3 shows the net benefit analysis of the QUIDS model (after recalibration of the intercept) compared to strategies of treat all (default) and treat none; the potential benefit (from reducing unnecessary treatment) and potential harms (from "missing" a case of preterm birth) are plotted on the same scale to allow direct comparison across a range of risk predictions. Alternative approaches of "Treating all" and "Treating none" are included for comparison. The QUIDS model is better than a treat-all approach at predicted risks of less than around 15%. The components of the receiver operator characteristic curve (sensitivity and 1-specificity) at different thresholds of risks are presented in Fig C in S1 Text. Using a risk threshold of 2% to define high risk, the model has sensitivity of 0.85 (95% CI 0.76 to 0.93) and specificity of 0.28 (95% CI 0.27 to 0.30).

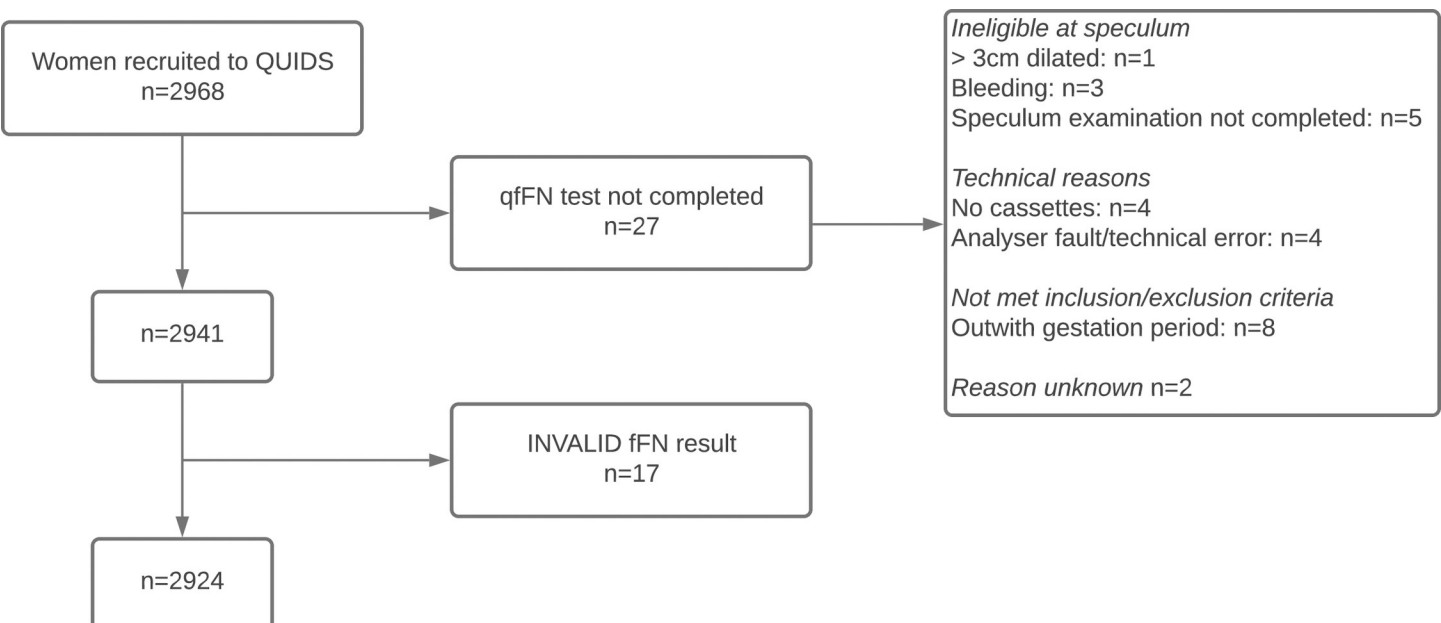

**Fig 1. Flowchart of participants in the QUIDS prospective cohort study.** Flowchart showing participants in the QUIDS prospective cohort study. fFN, fetal fibronectin; qfFN, quantitative fetal fibronectin.

## QUIDS risk prediction model

The externally validated QUIDS risk prediction model with intercept recalibrated for the prospective UK cohort population has the equation:

$$Log\ odds\ of\ spontaneous\ preterm\ birth\ within\ 7\ days\ of\ fFN\ test =$$

$$1.915*((qfFN + 1/100)^{0.5}) + 0.370*(Nulliparous) + 0.834*(Multiple\ pregnancy) -$$

$$0.687*(Current\ smoker) - 0.139*(Not\ White\ ethnicity) - 5.002,$$

where qfFN is the concentration of quantitative fFN in ng/ml, and Nulliparous, Multiple pregnancy, Current smoker, and Not White ethnicity are binary variables with a value of 0 or 1

**Table 3. Performance of QUIDS risk prediction model for the prediction of spontaneous preterm birth within 7 days of quantitative fFN test in external validation cohort.**

|  | A. Applying QUIDS model using pooled intercept from model development studies | B. Applying QUIDS model using updated intercept (recalibrated to UK population) |
|---|---|---|
| Intercept used in the model | −5.263 | −5.002 |
| **Model performance** |  |  |
| Nagelkerke $R^2$ | 36% (34%–38%) | 36% (33%–37%) |
| AUC | 0.89 (0.84–0.94) | 0.89 (0.84–0.94) |
| Expected/Observed | 0.80 (0.78–0.82) | 1 |
| Calibration-in-the-large | 0.26 (0.25–0.28) | 0 |
| Calibration slope | 1.24 (1.23–1.26) | 1.24 (1.23–1.26) |

Performance is shown when the intercept is derived from meta-analysis of the individual intercepts of the 5 contributing studies used for model development (A) and after recalibration updating the model intercept estimated using the external validation cohort (B).

AUC, area under the receiver : operating characteristic curve; fFN, fetal fibronectin.

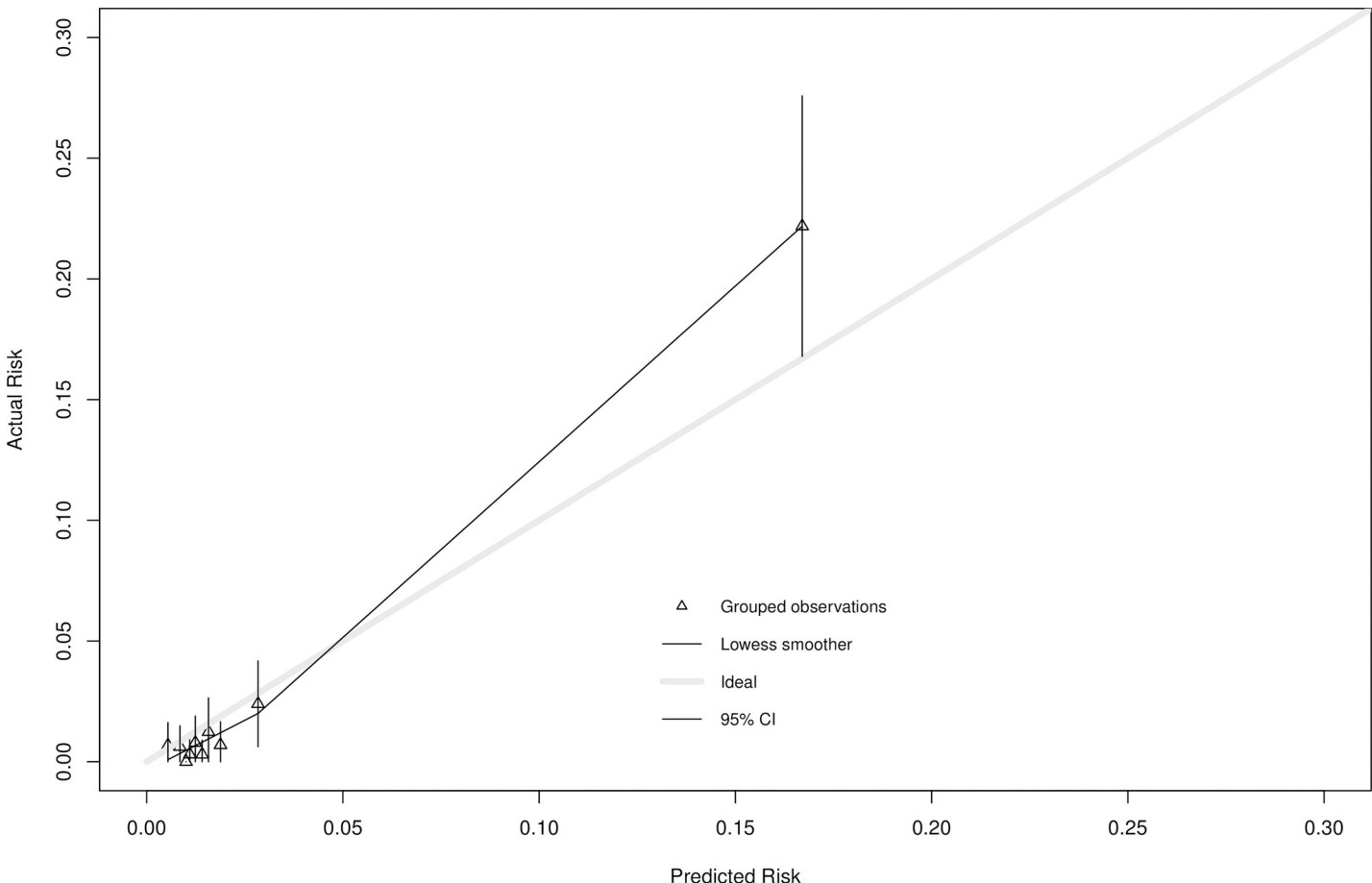

**Fig 2. Calibration plot of predicted versus observed risk for the QUIDS model validation in the prospective cohort data.** Calibration plot with predicted versus actual risk the QUIDS logistic regression model as applied to the prospective cohort study data (after multiple imputation; multiple imputation set 1 used to generate plot after recalibration with updated intercept). Data relating to grouped observations are shown in Table O in S1 Text. CI, confidence interval.

dependent on clinical risk factors. We present this as a risk prediction calculator that is available at https://argoshare.is.ed.ac.uk/connect/#/apps/340/access.

## Cost-effectiveness

Resource use and cost estimates from QUIDS prospective cohort study (Table Q in in S1 Text) were used in the final health economic model. The cost-effectiveness results comparing the QUIDS risk prediction model to a treat-all scenario and qualitative fFN are detailed in Table R in S1 Text (7-day time horizon) and Table S in S1 Text (lifetime horizon).

Compared with a treat-all strategy, the risk prediction model at ≥2% risk is associated with a reduction in QALDs of 0.0005 and a cost reduction of £866 over a 7-day horizon (incremental cost-effectiveness ratio [ICER]: £1,732,000, NMB: £856). Over a lifetime horizon, the risk prediction model at ≥2% risk is associated with a reduction in QALYs of 0.0006 and a cost reduction of £840 (ICER: £1,400,000, NMB: £827).

Compared to qualitative fFN alone, the risk prediction model increases costs by £41 per patient with a QALD gain of 0.002 over a 7-day horizon (ICER: £20,500, NMB: £-1). Over a lifetime horizon, the risk prediction model at ≥2% risk is associated with a QALY gain of

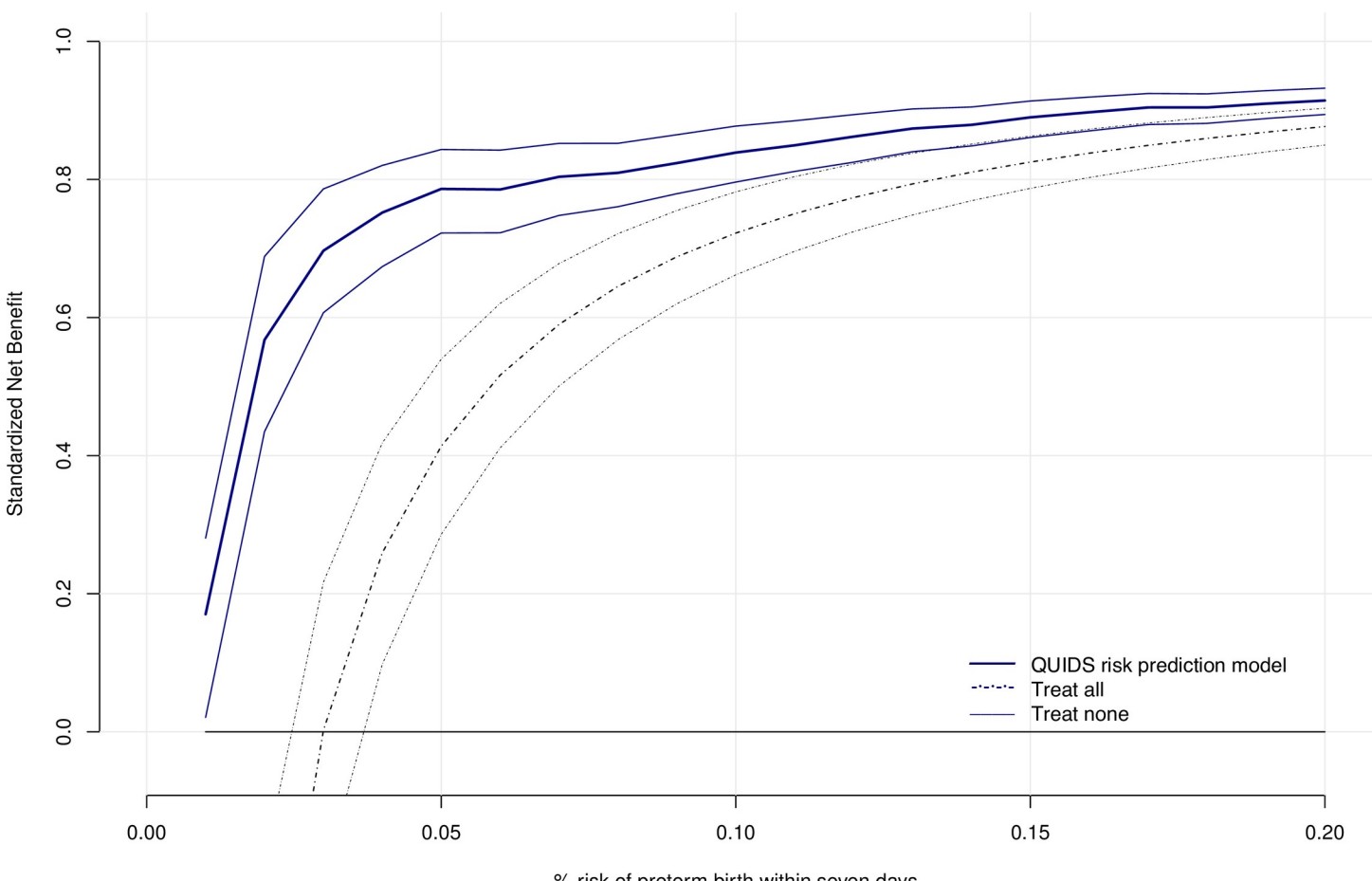

**Fig 3. Net benefit analysis of QUIDS risk prediction model.** Net benefit analysis (and 95% confidence intervals). The figure shows the standardised net benefit (to visualise the potential benefit from reducing unnecessary treatment, and potential harms from "missing" a case of preterm birth plotted on the same scale) at different % probabilities (from 0%–20%) of spontaneous preterm birth within 7 days as predicted by the QUIDS risk prediction model (blue lines) after recalibration of the intercept, compared to a policy of treating all women with symptoms (black dashed line) and a policy of treating no women (black horizontal line). Standardised net benefit was calculated using the formula $\text{sNB}^{(\text{opt}-\text{out})}(r) = \text{TNR}(r) - \left(\frac{\rho}{1-\rho}\right)\left(\frac{1-r}{r}\right)\text{FNR}(r)/p$, where sNB is defined by true negative rate (TNR), the false negative rate (FNR), the prevalence of preterm birth within 7 days (p), and the risk threshold (r) odds of low-risk designation at the % risk prediction [38]. FNR, false negative rate; sNB, standardised net benefit; TNR, true negative rate.

0.008 and an additional cost of £40 per patient (ICER: £5,000, NMB: £120). This is highly cost effective given the recommended NICE threshold of £20,000/QALY.

## Discussion

We have developed a model for risk prediction of preterm birth within 7 days of fFN test in women with signs and symptoms of preterm labour, based on the quantitative fFN test and clinical risk factors. The model shows promising performance when externally validated and recalibrated in a UK population, with potential net benefit when used to inform clinical decision-making, and is cost effective.

Current NICE guidance suggests that at gestations of 30 weeks or less, a treat-all strategy for women with signs and symptom of preterm labour is more cost effective than a strategy using tests of preterm labour [16]. Our cost-effectiveness analysis, based on prospectively collected data, found that the QUIDS risk prediction model at a risk prediction threshold of 2% was

likely to be cost effective compared to alternative strategies of treat-all, or use of qualitative fFN alone, across the range of gestations included in the study ($22+^0$ to $34+^6$ weeks gestation). Although the use of the QALY is limited in this context, we believe that a measure that attempts to capture the morbidity impact of missed treatment (false negatives) is crucial. However, further research is required regarding the measurement of quality of life for infants and how to appropriately value this.

Not all predictors in the QUIDS model performed as might have been predicted based on prior clinical knowledge. For example, smoking and not being of White ethnicity are both recognised to be risk factors for spontaneous preterm birth overall, but in our model, they were associated with a reduced risk of spontaneous preterm birth in women who present with symptoms. The reasons for this are unclear, but it may reflect an interaction between risk factors and management decisions. For example, clinicians' perception of smokers as being at "high risk" of preterm birth may mean they perform an fFN test more readily in the presence of minor symptoms in smokers than in nonsmokers. It should be noted that unexpected findings are a recognised phenomenon in risk prediction models [39]. As the main aim of a prediction model is to predict risk based on associations between predictors and the outcome, associations should not be interpreted causally, and individual predictor effects are, in principle, not relevant. Nevertheless, model face validity is helped if predictor effects are as expected [39], and further work will explore how best to present our risk prediction model and supporting information to clinicians to optimise confidence in the model.

Strengths of the study were size and methodological rigour, adhering to prespecified published protocols and reporting guidelines, and inclusion of sensitivity analyses. A potential limitation is the number of events in our external validation (85). We initially aimed to have at least 100 births within 7 days of testing, but the event rate was lower than anticipated, and we had a higher proportion of medically indicated preterm births than expected (prespecified as being excluded from our main analysis). Nevertheless, confidence intervals for the discrimination, calibration, and net benefit performance were narrow. Furthermore, to our knowledge, this is the largest published study of quantitative fFN, with more than 200 events included in the development and validation analyses in total. Another potential limitation is the amount of missing data for certain variables, for example, BMI in the development cohort. Multiple imputation was used to address missing data, which has been shown to be a valid technique for dealing with missing data within logistic regression models, resulting in less bias than excluding all women with missing data [27]. The majority of study participants were White, meaning that confidence intervals around risk estimates for women of other individual ethnic groups were large. We therefore collapsed ethnicity groupings into White and not White in our primary model. In future work, we would like to further refine risk estimates for women of different ethnicities. The cost-effectiveness analysis was carried out from the perspective of the UK NHS setting and may not be generalizable to other settings.

We believe that defining a single recommended risk prediction threshold to indicate treatment is undesirable as the relative weight of harms and benefits of treatment (including hospital admission) are likely to vary across individuals and healthcare settings, and the perception of benefits and harms are likely to be influenced by personal values and experience. We therefore propose that the QUIDS risk prediction model should be used to guide management decisions in the context of the woman's circumstances and preferences, as well as the healthcare setting. Nevertheless, relatively conservative and narrow-range risk prediction thresholds are indicated. In our prospective cohort study, the vast majority of preterm births within 7 days occurred in the highest decile of risk, but this equated to a predicted risk estimate of only around 20%. Furthermore, cost-effectiveness analysis supports use of a conservative risk threshold of 2%, and net benefit supported anything up to 15%.

Further evaluation of the risk prediction model in clinical practice is required to determine whether the risk prediction model improves clinical outcomes if used in practice. We recalibrated our model intercept to the UK population, as our original model was developed in an international set of studies. Further evaluation of this revised intercept, and whether recalibration is needed in each local centre, would be worthwhile.

In conclusion, the QUIDS risk prediction model including vaginal fluid quantitative fFN concentration and clinical risk factors showed promising performance in the prediction of spontaneous preterm birth within 7 days of testing and can be considered for use as part of a decision support tool to help guide management decisions for women with threatened preterm labour. It is readily implementable, with potential for immediate benefit to women and babies and health services, through avoidance of unnecessary admission and treatment.

## Supporting information

**S1 Table. Transparent Reporting of a multivariable prediction model for Individual Prognosis or Diagnosis (TRIPOD) [22] Checklist.**
(DOCX)

**S1 Text.** Table A. Details of inclusion and exclusion criteria for QUIDS study. QUIDS study inclusion and exclusion criteria. fFN, fetal fibronectin; IPD, individual participant data; RCT, randomised control trial. Table B. Prespecified candidate predictors for inclusion in the QUIDS model and availability in each included study. Candidate predictors were prespecified, based on their potential to influence risk of preterm birth, and included: fFN concentration (ng/ml), previous spontaneous preterm birth, nulliparity (no previous pregnancy >24 weeks), gestation at fFN test (weeks), maternal age (years), ethnicity, body mass index (BMI kg/m$^2$), smoking status, deprivation index, number of uterine contractions in set time period, cervical dilatation (cm), vaginal bleeding, previous cervical treatment for cervical intraepithelial neoplasia, cervical length (measured by transvaginal cervical length, mm), singleton or multiple pregnancy, and tocolysis. Only maternal age, BMI, ethnicity, smoking, nulliparity, multiple pregnancy, gestational age at assessment, previous spontaneous preterm birth before 34 weeks, cervical length, and fFN were available in each study, and, therefore, these 10 candidate predictors were included in the model development. Tocolysis was included in sensitivity analysis to explore any potential treatment effect on delaying birth. Table C. Details of eligible studies in the IPD meta-analysis. Six studies fulfilled the eligibility criteria; at the time, only one study was published (3), but three have subsequently been published (1, 2, 4). Five PIs agreed to provide data (Mol–Eufis; (2) van Baaren–Apostel-1; (1) Khalil–QFCAPS [Quantitative fetal fibronectin, Cervical length and ActimPartus for the prediction of Preterm birth in Symptomatic women]; unpublished, Shennan–EQUIPP [Evaluation of Fetal Fibronectin with a Quantitative Instrument for the Prediction of Preterm Birth]; (3) David–UCLH/Whit [University College London Hospital/Whittington], unpublished). The PI of the sixth study (STOP–Elovitz (4)) indicated data availability only after publication of the study, which occurred after completion of our analysis. fFN, fetal fibronectin. Table D. Assessment of bias of studies included in the IPD meta-analysis. Assessment of bias of studies included in the individual participant data meta-analysis. Risk of bias checklist adapted from (5). Table E. Results two-stage random effects meta-analysis for heterogeneity of predictor effects. Results of two-stage random effects meta-analysis for heterogeneity of predictor effects individual participant meta-analysis. fFN, fetal fibronectin. Table F. Sites for prospective cohort study. UK sites included in QUIDS prospective cohort study. Table G. Description of fFN test and protocol for sampling. Description of fetal fibronectin test and protocol for sampling. fFN, fetal fibronectin. Table H. Quality control measures for the Hologic Rapid fFN 10Q analyser. Description of quality control measures

for the fetal fibronectin test. fFN, fetal fibronectin. Table I. Sensitivity analyses for model development. Sensitivity analyses of different models developed using the IPD meta-analysis dataset. Multivariable logistic analysis of all candidate predictors; selected candidate predictors with cervical length (with data from the 3 studies with cervical length data completeness >80%); tocolysis. *qfFN = quantitative fetal fibronectin ((qfFN+1)/100)^0.5). Cervical length = ((cervical length+1)/10)^0.5).* Table J. Resource use items and unit collected via case report forms. Details of resource use items and unit collected via case report forms. *Hotel cost (58% of total stay cost) applied.* *Dosage based on British National Formulary recommended standard dosage (mg). Unit cost is then estimated by multiplying dosage received in mg by unit cost per mg.* C, cost. CPAP, Continuous Positive Airway Pressure therapy; LNU, Local Neonatal Unit; NICU, Neonatal Intensive Care Unit; SCBU, Special Care Baby Unit. Table K. Model parameters used in the cost-effectiveness analysis. Analysis: The cost-effectiveness of alternative risk prediction strategies was evaluated by its incremental cost-effectiveness ratio (ICER), which was calculated according to: ICER = ΔCosts/ΔQALY, where ΔCosts is the difference in total costs between risk prediction strategies, and ΔQALY is the difference in utility between risk prediction strategies. This incremental cost-effectiveness ratio can be compared against a societal willingness-to-pay for QALY gains (£20,000 in line with NICE reference case for cost per QALY. (15)) As considered QALYs a 7-day time horizon, we presented results in terms of cost per quality-adjusted life day (QALD), assuming a willingness-to-pay for QALD gained of £55 per day. The cost-effectiveness of the risk prediction strategies could also be converted to the NMB as there are multiple comparators. The NMB is a measure of the health benefit, expressed in monetary terms, which incorporates the cost of the new strategy, the health gain obtained, and the societal willingness to pay for health gains (£20,000). The NMB is expressed using the following formula: NMB = (E*WTP)–C. Where E = effectiveness; WTP = willingness-to-pay threshold; C = cost. The NMB approach is recommended when comparing more than one intervention and provides a clear decision rule (i.e., if NMB > 0, the new strategy is cost effective). Results can also be presented incrementally as the incremental NMB. Table L. Key assumptions for the cost-effectiveness models. Details of key assumptions made in the cost-effectiveness models. QALY, quality-adjusted life year. Table M. Baseline characteristics of the individual participant data meta-analysis dataset (A) and prospective cohort (B) study participants, derived from means of pooled imputations. Baseline characteristics of participants in QUIDS individual participant data meta-analysis dataset (A) and prospective cohort (B) study participants, derived from means of pooled imputations. fFN, fetal fibronectin; IQR, interquartile range; SD, standard deviation. Table N. Probability of spontaneous preterm birth within 7 days by gestational age in weeks. Table O. Data for calibration plot [Fig 2]. Mean expected and observed (with 95% confidence interval [CI]) risk of spontaneous preterm birth within 7 days for each of 10 risk groups (data for calibration plot [Fig 2]). Table P. Model performance measures for prespecified sensitivity analyses of the QUIDS model. Model performance measures for prespecified sensitivity analyses of the QUIDS model: Any preterm birth (i.e., including provider initiated preterm births); singletons only and complete case analyses. AUC, area under the curve. Table Q. Resource use and cost estimates from QUIDS prospective cohort. Resource use and cost estimates from QUIDS prospective cohort. CPAP, Continuous Positive Airway Pressure therapy; LNU, Local Neonatal Unit; NICU, Neonatal Intensive Care Unit; SCBU, Special Care Baby Unit. Table R. Cost-effectiveness results comparing the QUIDS risk predictor to a treat-all scenario and qualitative fFN (7-day time horizon). *ICER is in the southwest quadrant in cost-effectiveness plane (cost saved per QALD lost). ICERs above £20,000 are considered cost effective. QALD = Quality adjusted life days. ICER = Incremental cost-effectiveness ratio. fFN, fetal fibronectin; NMB, net monetary benefit. Table S. Cost-effectiveness results comparing the QUIDS risk predictor to a treat-all scenario and qualitative fFN

(lifetime horizon) *ICER is in the southwest quadrant in cost-effectiveness plane (cost saved per QALY lost). ICERs above £20,000 are considered cost-effective. fFN, fetal fibronectin; ICER, incremental cost-effectiveness ratio; NMB, net monetary benefit; QALY, quality-adjusted life years. Fig A. Decision tree framework. For each risk prediction strategy, the tree initiates with the true prevalence of preterm labour. Those identified as "high risk" by the strategy are admitted to a maternity unit with appropriate neonatal care facilities and receive antenatal corticosteroids. Those identified as "low risk" by the risk prediction strategy are not admitted and do not receive antenatal corticosteroids. The model can also be run under a hypothetical "treat all" strategy where all participants are admitted. The final destination in the decision tree is 1 of 5 possible states for preterm births: stillborn, minor morbidity, major morbidity, full health, and "did not give birth within 7 days." Given this structure, the model accounts for both the clinical and economic impact of false negative (low-risk result but gives birth within 7 days) and false positive results (high-risk result but does not give birth within 7 days) from the risk prediction strategies. False negatives represent missed opportunities to treat women with morbidity reducing antenatal corticosteroids, so infants born after a "false negative" result have a greater probability of experiencing neonatal morbidity and mortality, incurring the associated costs, quality of life, and survival impacts of these. False positives results will result in women being admitted to hospital unnecessarily, incurring additional and unnecessary cost of hospitalisation, interhospital transfer, and treatment. It is assumed that there are no quality of life side effects for receiving unnecessary treatment. Fig B. Net benefit curves comparing the QUIDS model with alternate models. The figure shows the standardised net benefit (to visualise the potential benefit from reducing unnecessary treatment, and potential harms from "missing" a case of preterm birth plotted on the same scale) at different % probabilities (from 0%–60%) of spontaneous preterm birth within 7 days as predicted by (i) the QUIDS risk prediction model including clinical risk factors and quantitative fetal fibronectin (fFN; blue line); compared to a policy of treating all women with symptoms (dark grey line); an alternate model including clinical risk factors and quantitative fFN + cervical length (red line); an alternate model including cervical length alone (green dashed line); and a policy of treating no women (light grey horizontal line).

Net benefit was calculated using the formula $\mathrm{sNB}^{(\mathrm{opt}-\mathrm{out})}(\mathrm{r}) = \mathrm{TNR}(\mathrm{r}) - \left(\frac{\rho}{1-\rho}\right)\left(\frac{1-\mathrm{r}}{\mathrm{r}}\right)\mathrm{FNR}(\mathrm{r})$,

where standardised net benefit (sNB) is defined by true negative rate (TNR), the false negative rate (FNR), the prevalence of preterm birth within 7 days (p), and the risk threshold (r) odds of low-risk designation at the % risk prediction [16]. Fig C. Plot of receiver operator curve components for QUIDS risk prediction model. Sensitivity (detection rate or true positive rate; black line) and 1-specificity (false positive rate; blue dashed line) and 95% CI at different % probabilities (between 0% and 10%) of spontaneous preterm birth within 7 days as predicted by the QUIDS risk prediction model. For example, at a predicted risk of 2%, the model has sensitivity of 0.85 (95% CI 0.76 to 0.93) and false positive rate of 0.28 (95% CI 0.27 to 0.30). The cost benefit axis is presented below the graph, with the cost indicating the ratio of "missed" cases (cost) to the number of cases where unnecessary treatment was avoided (benefit), at different levels of risk predicted by the QUIDS model.
(DOCX)

## Acknowledgments

We thank the members of the Trial Steering Committee (Philip Bennett [Chair], Melissa Whitworth, Olivia Wu, Peter Blair, Zing Gardiner, and Ben Wills) and study support staff (particularly the trial administrator Angela Niven, and latterly Rebecca Lees and Mary Paterson). We are grateful to Rita Sarqui and Foteini Emmanouella Bredaki who were coinvestigators on the

study from University College Hospital, London, which contributed data to the QUIDS IPD meta-analysis. We also thank Elspeth Horne for statistical advice and support and Laura Bonnett for support with R code.

We thank the researchers from each site who participated in the study: Dr Shona Cowan Royal Infirmary Edinburgh, Morag Dalton Royal Infirmary Edinburgh, Dr Alex Viner Borders, General Hospital, Dr Brian Magowan Borders General Hospital, Joy Dawson Borders General Hospital, Dr Shilpi Mittal University Hospital of Durham, Vicki Atkinson University Hospital of Durham Research, Jacqui Jennings Darlington Memorial Hospital, Dr Umo Essen South Tyneside District Hospital, Judith Ormonde South Tyneside District Hospital, Dr Vaideha Deshpande, Queen Elizabeth Hospital, Gateshead, Christine Moller-Christensen Queen Elizabeth Hospital, Gateshead, Phern Adams, Birmingham Women's Hospital, Nicola Farmer Birmingham Women's Hospital, Cody Allen Birmingham Women's Hospital, Dr Mani Malarselvi Birmingham Heartlands Hospital, Dr Lucy O'Leary Birmingham Heartlands Hospital, Lucy Sheppard Birmingham Heartlands Hospital, Dr Anurag Pinto Royal Gwent/Nevill Hall Hospital, Emma Mills Royal Gwent/Nevill Hall Hospital, Tracy James Royal Gwent/Nevill Hall Hospital, Kelly Griffiths Royal Gwent/Nevill Hall Hospital. Becky Westbury Royal Gwent/Nevill Hall Hospital, Patricia Jarvis Royal Gwent/Nevill Hall Hospital, Yaa Acheampong St Georges Hospital, Daniella Hake St Georges Hospital, Nessa Muhidun St Georges Hospital, Dr Jyothi Rajeswary Kings Mill Hospital, Katie Slack, Kings Mill Hospital, Caroline Moulds Kings Mill Hospital, Sarah Shelton Kings Mill Hospital, Mandy Gill Kings Mill Hospital, Dr Attila Vecsei St Richards and Worthing Hospital, Emma Meadows St Richards Hospital, Viv Cannons Worthing Hospital, Dr Sangeeta Pathak Hinchingbrooke Hospital, Tara Pauley Hinchingbrooke Hospital, Christie Oakes Hinchingbrooke Hospital, Kimberley Morris Hinchingbrooke Hospital, Charlotte Clayton Hinchingbrooke Hospital, Dr Marsham Moselhi Prince of Wales and Singleton Hospitals, Sharon Jones Prince of Wales and Singleton Hospitals, Helen Worrell Prince of Wales and Singleton Hospitals, Eve Watkins Prince of Wales and Singleton Hospitals, Maria Nash Prince of Wales and Singleton Hospitals, Sian Phillips Prince of Wales and Singleton Hospitals, Cath Jones Prince of Wales and Singleton Hospitals, Claire Vaughan Hughes Prince of Wales and Singleton Hospitals, Rhian Love Prince of Wales and Singleton Hospitals, Andrea Hill Prince of Wales and Singleton Hospitals, Rhian Lewis Prince of Wales and Singleton Hospitals, Dr Steve Wild University Hospital North Tees, Sharon Gowan University Hospital North Tees, Alison Samuels University Hospital North Tees, Dr Aparna Reddy Stoke Mandeville Hospital, Julie Tebbutt Stoke Mandeville Hospital, Dr Sarah Reynolds Bedford Hospital, Carina Gaplin Bedford Hospital, Marina Iaverdino Bedford Hospital, Dr Stewart Pringle Queen Elizabeth University Hospital Glasgow, Therese McSorely Queen Elizabeth University Hospital Glasgow, Kirsteen Paterson Queen Elizabeth University Hospital Glasgow, Dr Maheshwari Srinivasan Birmingham City Hospital, Sarah Potter Birmingham City Hospital, Sarah Figg Birmingham City Hospital, Lavinia Henry Birmingham City Hospital, Dr Matthew Hogg Royal London Hospital and Whipps Cross University Hospital, Zoi Vardavaki Royal London Hospital, Alice Rossi Royal London Hospital, Prudence Jones Whipps Cross University Hospital, Dr Sujatha Thamban Whipps Cross University Hospital, Dr Saumitra Sengupta Queen Alexandra Hospital Portsmouth, Zoe Garner Queen Alexandra Hospital Portsmouth, Amanda Hungate Queen Alexandra Hospital Portsmouth, Berni Edge Queen Alexandra Hospital Portsmouth, Layla Toomer Queen Alexandra Hospital Portsmouth, Kay Andrews Queen Alexandra Hospital Portsmouth, Faith Hagger Queen Alexandra Hospital Portsmouth, Dr Chineze Otigbah Queens Hospital Romford, Anne-Marie McGregor Queens Hospital Romford, Elsie Uwegba-Obatarhe Queens Hospital Romford, Dr Chandrima Biswas Whittington NHS Trust, and Dr Ora Jesner Whittington NHS Trust. We also thank the wider clinical team and all others who contributed to the screening, recruitment, and

outcome data collection, as well as the R&D offices at participating sites, and in particular all the women who participated in QUIDS.

The views expressed are those of the authors and not necessarily those of the NHS, the NIHR, or the Department of Health and Social Care.

## Author Contributions

**Conceptualization:** Sarah J. Stock, Kathleen A. Boyd, Manju Chandiramani, Ben W. Mol, Jane E. Norman.

**Formal analysis:** Margaret Horne, Merel Bruijn, Helen White, Kathleen A. Boyd, Robert Heggie, Lorna Aucott, Ewoud Schuit, Richard D. Riley, John Norrie.

**Funding acquisition:** Sarah J. Stock, Kathleen A. Boyd, Rachel K. Morris, Jon Dorling, Lesley Jackson, Asma Khalil, Andrew Shennan, Tina Lavender, Susan Harper-Clarke, Ben W. Mol, Richard D. Riley, Jane E. Norman, John Norrie.

**Investigation:** Sarah J. Stock, Merel Bruijn, Helen White, Kathleen A. Boyd, Rachel K. Morris, Anna L. David, Asma Khalil, Andrew Shennan, Gert-Jan van Baaren, Victoria Hodgetts-Morton, Ewoud Schuit.

**Methodology:** Sarah J. Stock, Merel Bruijn, Kathleen A. Boyd, Lorna Aucott, Rachel K. Morris, Jon Dorling, Lesley Jackson, Anna L. David, Andrew Shennan, Tina Lavender, Ewoud Schuit, Susan Harper-Clarke, Ben W. Mol, Richard D. Riley, John Norrie.

**Project administration:** Lisa Wotherspoon.

**Supervision:** Sarah J. Stock, Richard D. Riley, Jane E. Norman, John Norrie.

**Writing – original draft:** Sarah J. Stock.

**Writing – review & editing:** Margaret Horne, Merel Bruijn, Helen White, Kathleen A. Boyd, Robert Heggie, Lisa Wotherspoon, Lorna Aucott, Rachel K. Morris, Jon Dorling, Lesley Jackson, Anna L. David, Asma Khalil, Andrew Shennan, Gert-Jan van Baaren, Victoria Hodgetts-Morton, Tina Lavender, Ewoud Schuit, Susan Harper-Clarke, Ben W. Mol, Richard D. Riley, Jane E. Norman, John Norrie.

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
