## [Editor Report · Decision Letter 0]

10 Mar 2021

Dear Dr Stock, 

Thank you for submitting your manuscript entitled "Development and validation of a risk prediction model of preterm birth for women with preterm labour symptoms (the QUIDS study): individual participant data meta-analysis and prospective cohort study." for consideration by PLOS Medicine.

Your manuscript has now been evaluated by the PLOS Medicine editorial staff and I am writing to let you know that we would like to send your submission out for external assessment.

Once your full submission is complete, your paper will undergo a series of checks in preparation for external assessment. 

Kind regards,

Richard Turner, PhD

rturner@plos.org

---

## [Decision Letter · Decision Letter 1]

17 Apr 2021

Dear Dr. Stock,

Thank you very much for submitting your manuscript "Development and validation of a risk prediction model of preterm birth for women with preterm labour symptoms (the QUIDS study): individual participant data meta-analysis and prospective cohort study." (PMEDICINE-D-21-01141R1) for consideration at PLOS Medicine. 

Your paper was discussed with an academic editor with relevant expertise and sent to independent reviewers, including a statistical reviewer. The reviews are appended at the bottom of this email and any accompanying reviewer attachments can be seen via the link below:

[LINK]

In light of these reviews, we will not be able to accept the manuscript for publication in the journal in its current form, but we would like to invite you to submit a revised version that addresses the reviewers' and editors' comments fully. You will recognize that we cannot make a decision about publication until we have seen the revised manuscript and your response, and we expect to seek re-review by one or more of the reviewers. 

We hope to receive your revised manuscript by May 10 2021 11:59PM. Please email us (plosmedicine@plos.org) if you have any questions or concerns.

Please let me know if you have any questions, and we look forward to receiving your revised manuscript. 

Sincerely,

Richard Turner, PhD

rturner@plos.org

Please add "SJS is a member of PLOS Medicine's Editorial Board" to the competing interest statement. 

We suggest adapting the title to "...: A prospective cohort study and individual participant data meta-analysis".

Please trim the "Background" subsection of your abstract.

Please quote aggregate participant demographic details in the abstract.

Please add a new final sentence to the "Methods and findings" subsection of your abstract, which should begin "Study limitations include ..." or similar and quote 2-3 of the study's main limitations. 

After the abstract, please add a new and accessible "author summary" section in non-identical prose. You may find it helpful to consult one or more published research papers in PLOS Medicine to get a sense of the preferred style. 

Please highlight any analyses that were not prespecified. 

In the Discussion section, we suggest expanding the discussion of study limitations. 

Please substitute "White" for "Caucasian".

Please avoid claims such as "the largest", and if used add "to our knowledge". 

Generally the style should be "We included 10 predictors ...", for example, although numbers should be written out at the start of sentences. 

Throughout the paper, please adapt reference call-outs to the following style: "... every year [1,2]." (noting the absence of spaces within the square brackets). 

In the reference list, please abbreviate journal names consistently. Please convert all italics to plain text. 

Please break the attached TRIPOD checklist out into a separate file, labelled "S1_TRIPOD_Checklist" or similar and referred to as such in the Methods section. 

In the checklist, individual items should be referred to by section (e.g., "Methods") and paragraph numbers rather than by line or page numbers, as the latter generally change in the event of publication. 

Comments from the reviewers:

*** Reviewer #1: 

This study evaluated the value of quantitative fFN for prediction of imminent delivery among symptomatic women with threatened preterm labor. The authors proposed a new risk model and evaluated cost-effectiveness. Overall the study addresses an important problem and the methodology is sound. 

1) It the introduction, the authors could perhaps mention studies aiming at developing signatures for prediction of preterm birth in symptomatic patients, such as https://pubmed.ncbi.nlm.nih.gov/24828675/ and other.

2) In the introduction, the authors state: "Transvaginal ultrasound measurement of cervical length is one strategy which has moderate predictive performance, but it requires specialised equipment and highly trained staff."

The authors could expand this sentence to add that there are personalized methods to assess cervical length which leads to better prediction including in symptomatic women (https://pubmed.ncbi.nlm.nih.gov/32918893/).

3) Table 1 is said to show "Baseline characteristics of the IPD meta-analysis dataset (A) and prospective cohort (B) study participants, derived from means of pooled imputations". Superscripts could be used to specify for each variable how many cases were missing and hence needed imputation.

4) In Table 2 and line 440, smoking and Non-Caucasian Ethnicity seem have a protective effect against delivery within 7 days in symptomatic women. Perhaps these findings could be discussed in the context of what is known.

5) The authors state "Quantitative fFN was transformed (square root) because of non-linearity", yet the formula in Table 2 "((qfFN+1)/100)^-0·5)" shows an inverse square root (-0.5). This is unlike in line 441 where square root (0.5) is used instead. Please clarify which is the right one. 

6) "Candidate predictors (i.e. those, considered to have with potential to influence"

Remove "with"

*** Reviewer #2: 

"Development and validation of a risk prediction model of preterm birth for women with preterm labour symptoms (the QUIDS study): individual participant data meta-analysis and prospective cohort study" describes the development and validation of a model for predicting preterm birth (< 37 weeks) as its outcome, using vaginal fluid fetal fibronectin (quantitative fFN) as the primary novel risk factor/predictor of interest, together with nine other common predictors (reduced to five in total during development for the proposed QUIDS, formula in Line 440). Logistic regression with random effects meta-analysis was used as the underlying model, and compared to models including all potential predictors, and with cervical length, with all models returning similar performance. External validation and calibration was performed on a prospective cohort of over 2,900 patients, with consistent performance before and after calibration. A brief cost-effectiveness analysis supporting use of QUIDS by assessing reduction in QALDs and monetary costs was provided, and suggests that usage of QUIDS at a suitable risk prediction threshold (2%; sensitivity 0.79, specificity 0.84) may be superior to existing treat-all or qualitative fFN-only approaches.

The study is generally meticulously documented, much of it in the accompanying supporting information. However, there remain some concerns that might be clarified:

1. The qualifying signs and symptoms of preterm labour (Line 163) applied in this study, might be described in greater detail.

2. It is notable that despite "gestational age at assessment" being one of the original ten candidate predictors, it was not judged sufficiently useful to be included in the final QUIDS model, despite patients' values for gestational age ranging from 26 to 32 weeks in the development dataset (Table 1). In particular, one might expect (premature) delivery at say 31 weeks, to be more common than at say 27 weeks, other conditions being equal. It might be discussed as to whether preterm births closer to 32 weeks are indeed more common from the available data, or if the presence of preterm labout symptoms overrides such expectations; the baseline probability of (preterm) delivery from gestational age might also be described.

3. While a main focus of the paper is the inclusion of qfFN with other clinical predictors (with models involving all predictors/cervical length also examined), it does not appear to be established that the proposed QUIDS also significantly outpeforms yet more-parsimonious models (in particular, qfFN alone, or say qfFN+multiple pregnancy+current smoker for top three highest predictor weights); the backwards selection procedure for selecting predictors from the candidate set might also be described in greater detail.

4. Given the focus on qfFN in the study, it might be considered to provide more information about the distribution of qfFN (quantitative, ng/ml) depending on outcome, perhaps as a forest plot. This might also be considered for the other QUIDS predictors.

5. For the dataset demographics (as provided in Table 1), a confusion matrix relating major predictor "Qualitative fFN positive" and outcome "Spontaneous preterm birth within seven days" might additionally be presented (i.e. for each dataset, these correspond to the four values "qfFN positive/preterm birth", "qfFN positive/no preterm birth", "qfFN negative/preterm birth", "qfFN negative/no preterm birth")

6. For Figure 2, the data corresponding to each grouped observation point might be described (possibly in supporting information) and labelled.

7. For Figure 3, the various lines might be labelled as to their respective probabilities, for each category.

Minor issues:

8. In Line 125, it is asserted that "The most established biochemical test is vaginal fluid fetal Fibronectin". This assertion might be directly supported by citation(s) if possible.

9. In Line 367, "internally validation" -> "internal validation"

*** Reviewer #3: 

Thank you for letting me review this paper about a new prediction model for preterm birth in symptomatic women. This paper is about combining clinical risk factors with quantitative fetal fibronectin in a risk prediction model for women presenting with signs and symptoms of preterm birth. The study is well designed with a development dataset and a validation dataset and I congratulate the authors for a great achievement. I have a few comments and suggestions for the manuscript. 

Overall comments

1) I understand that the common practice in the UK is to use the qualitative fibronectin test rather than cervical length. Though, since a majority of countries, at least in Europe, use cervical length screening for preterm birth in symptomatic women, I would have liked to see the comparison of the predictive accuracy not only in comparison to qualitative fibronectin but also to cervical length screening with the best cut-off to the specific population. From the data, I understand that cervical length was not measured in all studies in the development dataset but would it be possible to use the studies where cervical length was measured, include the absolute numbers of cervical length in mm and provide data also for the predictive accuracy of cervical length only both in the development and validation dataset (if cervical length is available in the validation dataset)? If so, it would be of interest to readers in countries using cervical length screening to be able to compare current practice to the prediction model.

2) The comparison of the new model to the standard of care with "treat all" below 30 gw and according to positive FfN above 30 gw would proabably not be the routine in other countries where a more restrictive approach below 30 gw would be more common (for example with a normal cervical length where the woman would be re-assessed later on). Would this change the cost effectiveness analysis if comparing the new model to a more conservative approach? 

Specific comments

METHODS

1) Line 155; more useful predictor of PTB - compared to what? Dichotomous test?

2) Line 167: One of the included papers had an outcome of PTB<34 weeks, not days to delivery. Did the author provide data so that thus could be calculated?

3) Line 168: an extra "with" could be removed (have with potential...)

4) Line 173; change was to were

5) Line 280: When adding iatrogenic preterm birth in the outcome - what is the underlying rationale to why fibronectin would be increased before iatrogenic PTB? Please explain.

6) Line 296-304: Did you also considered a machine learning based approach to allow for non-linear correlations in the data?

7) Line 301: How did it become apparent that the incidence of PTB in your population was lower than expected? Did you investigate this specifically in the dataset and was that a planned analysis?

RESULTS

8) Table 1; Do you have any explanation to why the incidence of PTB was smaller than expected in your cohort? How was the external validity of your population? Did you hold a record of women declining participation or was there any differences in the population of the included centers in your study compared to other UK centers?

9) Why do you think BMI did not fall out as a covariate after backwards selection when it is a known risk factor for PTB? Could it have to do with large missing data for BMI?

10) Line 370: Here you state that the model with cervical length performed better with an AUC of 0.92 but had similar potential utility? Again, referring to the overall comment, it would be interesting, taking that many countries use cervical length, to compare this to FfN and also in the final model for validation if possible. 

11) Lines 417-418: In this analysis you seem to have compared the new model to "treat all" or "treat none". I understood from earlier in the manuscript that "treat all" was only below 30 gw. Could you please explain?

12) Lines 440-442: The prevalence of smoking was high both in the development and in the validation cohort. Is this representative for the UK population / other populations with women with symptoms of PTB?

13) Cost effectiveness: How is it possible to understand the potential benefits of timely MgSO4, transfer to a tertiary center and timely corticosteroids compared to the potential harm of repeated corticosteroids / over-treatment? Is this an analysis that we can actually relate to in clinical practice? I find it difficult to assess. Also, here it is stated that it was compared to a "treat all" strategy (see question 11).

DISCUSSION

If not possible to add a comparison to CL only and to validate a prediction model with CL instead of FfN, it would be nice to see a section in the discussion adding this possibility for countries where CL is well implemented already and available day and night.

***

[LINK]

---

## [Decision Letter · Decision Letter 2]

25 May 2021

Dear Dr. Stock,

Thank you very much for re-submitting your manuscript "Development and validation of a risk prediction model of preterm birth for women with preterm labour symptoms (the QUIDS study): A prospective cohort study and individual participant data meta-analysis" (PMEDICINE-D-21-01141R2) for review by PLOS Medicine.

I have discussed the paper with my colleagues and the academic editor and it was also seen again by one reviewer. I am pleased to say that provided the remaining editorial and production issues are dealt with we are planning to accept the paper for publication in the journal.

[LINK]

We look forward to receiving the revised manuscript by Jun 01 2021 11:59PM.   

Sincerely,

Louise Gaynor-Brook, MBBS PhD

Associate Editor 

PLOS Medicine

plosmedicine.org

Requests from Editors:

Data availability:

Thank you for making your data available without restrictions on access, in a public repository. Prior to publication of your manuscript, please do ensure to provide us with the DOI or accession number required in order for this to be accessed. Could you please also confirm whether the data included in your meta-analysis (including the unpublished data from Khalil et al and David et al) will be available in the same public repository?

Abstract Methods and Findings:

For the meta-analysis component of your study, please provide the years during which the prospective cohort studies analysed took place. 

For the prospective cohort study component of your study, please provide the years during which the study took place. 

Please begin your Abstract Conclusions with "In this study, we observed ..." or similar, to summarise the main findings of your study. 

Thank you for providing an Author Summary. In the final sentence of ‘What do these findings mean?’, please also provide a bullet point summarising the main limitations of your study that will be accessible to a wide audience. 

Line 118 - please clarify what is meant by ‘ and the risk predictor generates’

Line 121 - please add the source of the data in each case (European / UK)

Introduction:

Line 151 - please provide a couple of examples of the adverse maternal and neonatal effects

Please indicate whether your study is novel, ensuring to temper assertions of primacy with ‘to the best of our knowledge’ or similar

Please remove the final line ( indicating your study results and conclusion) from your Introduction.

Results: 

Please incorporate Tables S13 and S15 into the main text of your results, as the baseline characteristics of study participants (without imputation)

Line 378 - Please provide the years during which the prospective cohort studies analysed took place. 

Discussion:

Please re-organize the Discussion a little, as follows: a short, clear summary of the article's findings; what the study adds to existing research and where and why the results may differ from previous research; strengths and limitations of the study; implications and next steps for research, clinical practice, and/or public policy; one-paragraph conclusion.

Line 519 - please revise the sentence beginning ‘ Further, to our knowledge the largest published study...‘ as the meaning is not entirely clear 

Figures and Tables:

Please provide titles and legends for each individual table and figure

Please define all abbreviations used in each figure in the respective figure legend e.g. Fig 1 “qfFN”

Please define all abbreviations used in each table in the respective table legend e.g. Table 1 “fFN”

References:

Please ensure that journal name abbreviations match those found in the National Center for Biotechnology Information (NCBI) databases, and are appropriately formatted and capitalised. 

Please remove additional information such as ‘[published Online First: 2018/11/06]’ in ref 1

Ref 15 - please provide a valid link as the text appears to have merged

Supplementary files: 

Please provide titles and legends for each individual table and figure in the Supporting Information. Please also define all abbreviations used in each table/figure in the respective table/figure legend. 

Thank you for providing the TRIPOD checklist; for points 21 and 22, please provide the (sub)sections within the manuscript text where this information can be found

Comments from Reviewers:

Reviewer #2: We thank the authors for addressing our previous points, and have no further comments.

[LINK]

---

## [Editor Report · Decision Letter 3]

4 Jun 2021

Dear Dr. Stock,

Thank you very much for re-submitting your manuscript "Development and validation of a risk prediction model of preterm birth for women with preterm labour symptoms (the QUIDS study): A prospective cohort study and individual participant data meta-analysis" (PMEDICINE-D-21-01141R3) for review by PLOS Medicine.

The manuscript looks almost ready for editorial acceptance, but I wondered if I could ask you please to correct Table S13 (the table seems to have been pasted twice) and to remove both instances of the word 'new' on lines 175 and line 520? 

As usual, we would expect to receive your revised manuscript within 1 week, but please email me (lgaynor@plos.org) if that isn't possible.

We look forward to receiving the revised manuscript by Jun 11 2021 11:59PM.   

Best wishes,

Louise Gaynor-Brook, MBBS PhD

Associate Editor 

PLOS Medicine

plosmedicine.org

[LINK]

---

## [Editor Report · Decision Letter 4]

7 Jun 2021

Dear Dr Stock, 

On behalf of my colleagues and the Academic Editor, Prof. Gordon Smith, I am pleased to inform you that we have agreed to publish your manuscript "Development and validation of a risk prediction model of preterm birth for women with preterm labour symptoms (the QUIDS study): A prospective cohort study and individual participant data meta-analysis" (PMEDICINE-D-21-01141R4) in PLOS Medicine.

PRESS

Sincerely, 

Louise Gaynor-Brook, MBBS PhD 

Associate Editor 

PLOS Medicine